# Blended and collaborative learning: Case of a multicultural graduate classroom in Taiwan

**Aurora V. Lacaste**[1,2]**, Ming-Min Cheng**[3]**, Hsueh-Hua Chuang**[1,3]*

**1** International Graduate Program of Education and Human Development, National Sun Yat-Sen University, Kaohsiung, Taiwan, **2** University of the Philippines Open University, Laguna, Philippines, **3** Institute of Education, National Sun Yat-Sen University, Kaohsiung, Taiwan

* hsuehhua@g-mail.nsysu.edu.tw

**Data Availability Statement:** All relevant data are within the manuscript and its Supporting information files.

**Funding:** This work was supported by Office of Higher Education Sprout Project 2020 from

## Abstract

As internationalization of higher education has become more prevalent, the search for approaches to support quality learning within multicultural classrooms has become critical. In this study, we presented a blended and collaborative learning (BCL) method and discussed its potential as a viable learning approach for graduate classrooms with culturally-diverse students. We first discussed implementation of a BCL approach in a multicultural graduate course, then explored learnings gained from the BCL experience in terms of three interdependent Community of Inquiry elements: teaching, cognitive, and social presences, and in terms of its implementation in a multicultural setting.

## Introduction

As Taiwanese universities along with other Asian universities adopt internationalization in seeking global competitiveness, the need to address the academic and social challenges of culturally-diverse students—both international and domestic—becomes both urgent and necessary.

Studies on the factors affecting the successful realization of internationalization in Taiwan usually focus on language policy [1, 2]; disposition of international students, including their in-class behaviors, perceptions, and norms [3–5]; and curriculum development and improvement [6, 7]. Although these studies have led to a better understanding of what must be done to address the challenges related to internationalization at the national and institutional levels, to understand the challenges multiculturalism brings at the teaching and learning levels, there remains a need to conduct research on what happens within classrooms and to address those challenges by providing and implementing alternative solutions.

When students first enter a multicultural classroom, along with attaining the usual goal of meeting the learning objectives of the course, they must adjust to a "new cultural environment," navigating unfamiliar people and cultures [8]. This makes learning in multicultural classrooms more challenging for both teachers and students; therefore, a search for teaching approaches and optimal conditions to enhance learning becomes necessary.

In this study, we described a blended and collaborative learning (BCL) approach to a multicultural graduate course, and examined students' perceptions about the course. Our goal is to

Ministry of Education in Taiwan. (no grant number available). The funder had no role in the study design, data collection and analysis, decision to publish, or preparation of the manuscript.

**Competing interests:** The authors have declared that no competing interests exist.

present a case and enumerate learnings on the use of both blended and collaborative learning activities leading for active and engaged learning in multicultural graduate classrooms.

The purpose of this study is two-fold. First, we aimed to demonstrate the implementation of a BCL course to encourage cooperative learning among graduate students of different cultural backgrounds. We adopted the framework for blended learning design proposed by Neumeier [9] to present the findings. We also described how technology, online and offline instruction, and learning activities were combined to implement active learning methods such as collaborative learning. Second, we used the community of inquiry (CoI) framework [10] to discover the features of a BCL environment that are perceived by students as being associated with learning and satisfaction.

## Theoretical background

### Multicultural classrooms in Taiwan

In this study, we considered a classroom to be multicultural if it included at least five individuals from a minority group (i.e., individuals who were born in a country different from their country of residence or whose parents are from other countries) or was represented by at least two different cultural groups [11].

Multicultural classrooms in Taiwan are becoming increasingly common owing to the large number of international students who choose to study in Taiwan, which increased by 84.78% in 2019 [12]. Besides overseas Chinese students, most graduate students came from Southeast Asia and South Asia as a result of the New Southbound Policy that provided scholarships for students from those regions [12]. According to DeAeth [13], because the foreign students enrolled in 2018 accounted for about 10% of the total university and college students enrolled in Taiwan, a multicultural classroom in Taiwan could be described as that comprising of both local students and international students from other Asian countries.

Because the educational beliefs and practices among local and international students are diverse and their perceptions of cultural distance vary [14], multicultural classrooms are not without challenges. Such challenges brought about by cultural diversity in the classroom may inhibit the achievement of some of the main aims of internationalization: to allow culturally diverse students to form a "cultural mix" and to develop cross-cultural relationships while learning [14, 15]. Therefore, developing intervention programs, such as teaching and learning approaches that could promote social cohesion and meaningful learning, and subsequently assessing their impact is important.

### Blended and collaborative learning

Blended learning, a learning experience that integrates face-to-face and online instruction [16], is an innovative practice adopted by many higher education institutions since the inception of technology-based training in the 1960s. A blended environment is said to bring continuality of learning experiences from multiple times and spaces to create more holistic and integrated learning experiences [17]. Blended learning is seen as a promising teaching and learning principle in many higher education institutions, as it helps achieve learning by combining technologies with face-to-face learning, thus maintaining regular interactions among students and teachers [18]. Collaborative learning, in contrast, is an instructional approach involving a small group of students interacting with one another and sharing their knowledge and skills to reach specific learning goals [19]. We define BCL as learning through collaborative activities while using two modes of communication or instruction (i.e., face-to-face and online modes). So and Bonk [17] described the roles of blended learning approaches in

collaborative learning environments. In our study, we extended this investigation of blended learning in collaborative environments in a multicultural setting.

Rationales for adopting blended learning are varied and occur at different levels. At the institutional level, the rationales include flexibility of provision, support for diversity through equity of access among students, enhancement in learning experience, capability to operate in a global context, and efficiency [20]. At the course level, blended learning has been adopted as a response to practical challenges faced by faculty handling a course, including staff-student contact and pedagogical effectiveness [20, 21].

There is no general definition of blended learning, as researchers and institutions differ in limiting the definition boundaries; some emphasize the overall reduction in face-to-face contact time, whereas others specify the levels of online and face-to-face instruction [16, 22, 23]. Stacey and Gerbic [24] recognized that blended learning can be positioned somewhere between fully online and fully face-to-face courses, although definitional issues exist with respect to where blended learning might be in this continuum. Garrison and Kanuka [25] believed that the real test of blended learning is how effectively the face-to-face and online components are integrated.

The literature on blended learning elucidates the favorable results produced by such an approach to student success and satisfaction at the course level and to the generation of a strong sense of community among students [20, 26, 27].

This study used Neumeier's [9] framework of parameters for the design of blended learning in examining a blended learning approach implemented in a multicultural graduate class; we adopted this particular framework to clarify the details and structure of the blended learning environment described in this study. Neumeier's framework provides a systematic way of describing blended learning environments that focuses on the requirements specific to the teaching and learning of a particular subject. The original framework has six parameters for designing blended learning; although its development was based on language teaching and learning, it can very well be adapted and applied to blended learning in other courses. The adapted parameters for this study include (a) mode, (b) model of integration, (c) distribution of learning content and objectives, (d) teaching methods (originally language-teaching methods), (e) involvement of learning subjects, and (f) location.

Collaborative learning is a learner-centered approach to teaching and learning that involves groups of students working together to perform a given task [28]. According to Vygotsky [29], individuals can achieve higher levels of learning through collaborative learning, during which individuals adjust their understanding of knowledge concepts by sharing experiences and exchanging information and insights. Studies have found that collaborative learning has an advantage over individual learning because it supports sharing with others new knowledge that can be reorganized alongside old knowledge [30]. Technology can be effectively integrated into classrooms [31], offering students increased opportunities to engage in group tasks, including sharing of sources, interaction, collaborative learning, and building communities [32, 33]. Chang et al. [34] demonstrated how technology-mediated online collaboration can increase a student's engagement and learning proficiency.

Tielman et al. [11] studied the practices and perceptions of both teachers and students in a vocational secondary school in the Netherlands with respect to collaborative learning in a multicultural class. They reported poor collaboration in the multicultural class, as students focused only on their own personal activities. The teachers, too, seemed unaware of their roles in affecting students' behaviors and the influences of students' cultural backgrounds in the collaborative process. Le, Janssen, and Wubbels [35] pointed out that students, in general, can encounter difficulties in collaborative learning if they lack collaborative skills such as accepting opposing viewpoints, giving elaborate explanations, providing and receiving help, and

negotiating. In a multicultural classroom, where students come from different cultural backgrounds, it could be more challenging to use these collaborative skills. However, Coelho ([36], as cited in [11]) argued that collaborative learning is an appropriate teaching approach in the multicultural context. Gay [37] and Sharan [38] also emphasized the role of teachers in creating a culturally sensitive classroom by being aware of students' diverse cultural backgrounds and integrating cooperative learning methods [38]. Developing a rich repertoire of multicultural instructional examples when teaching school subjects is important for teachers [37]. Sharan [38] suggested the use of cooperative learning methods and strategies but stressed that only placing students into groups is not sufficient; a partnership needs to be created in a multicultural classroom by the judicious and gradual use of cooperative learning methods with sensitivity to the manifestations of diversity. Moreover, Kumi-Yeboah et al. [28] found that minority graduate students perceived online collaborative learning as providing them with opportunities to share and lead discussions with different perspectives.

Studies on blended learning and collaborative learning are among the most cited works, as evidenced in reviews in educational technology; this indicates the significance of BCL as a current and potentially competitive research area. Blended learning, for instance, started receiving increased attention from the academic community since 2003–2006, showing an increasing interest in teaching and learning using technologies and games [39]. The topic of context and collaborative learning is one of the most studied topics, demonstrating its popularity in teaching and learning research, and its importance in promoting the development of various types of knowledge [39].

Although certain aspects of blended learning and collaborative learning have been previously studied, research describing what actually happens in a blended and collaborative environment and what lessons might be learned from such a study, particularly in a multicultural higher education classroom, is limited. Empirical studies with similar goals and format to the present study have been published before, signaling the importance of reporting the design, implementation, and evaluation of learner-centered educational initiatives in the digital era. An example is a study about an artificial intelligence (AI) literacy course attended by Hong Kong university students who are from diverse backgrounds [40]. The design, implementation, and assessment of the AI course with a flipped learning classroom approach were reported. Quantitative analysis was used to evaluate the course's impact to students' conceptual understanding [40].

## Community of inquiry (CoI)

Blended learning facilitates the development of a community of inquiry (CoI) [25, 41], referring to a group of individuals, particularly instructors and students, who collaboratively engage in critical discourse and reflection to create personal meaning and confirm mutual understanding [10, 42]. The CoI framework represents a process of creating a deep and meaningful educational experience through three interdependent elements: social presence, cognitive presence, and teaching presence [10].

Social presence is the ability of the participants to identify with the community, purposefully communicate in a trusting environment, and develop interpersonal relationships by way of projecting their individual personalities [10]. Studies have found that social presence has a significant relationship with both teaching presence and cognitive presence. Tolu [43] pointed out that social presence should be sustained through both synchronous and asynchronous course design elements, including instructor availability, collaborative activities, prompt feedback, and group formation. Zhang [44] further proposed that social presence supports learning and may lead to successful cognition.

Richardson et al. [45] stated that course design should include activities that encourage and help interaction, build trust among students, and allow students to participate in discussion [45]; these are evident features of a learning environment in which students have a sense of belonging within the class, can interact and openly discuss with one another, and can identify themselves as members of the group. Richardson et al. [45] further proposed that problem-solving tasks, collaborative projects, and small-group discussions may be beneficial in building group cohesion.

Teaching presence includes the design, facilitation, and direction of cognitive and social processes for the purpose of realizing personally meaningful and educationally worthwhile learning outcomes [10]. Teaching presence has three characteristics: design and organization, facilitating discourse, and direct instruction. Previous research has emphasized the importance of teaching presence in an online learning environment [46]. Note that instructors can facilitate discussion and provide feedback that promote student learning while students are actively developing metacognitive awareness to manage their learning [47].

Cognitive presence, mediated by teaching and social presences within a collaborative experience [48], is the extent to which learners are able to construct and confirm meaning through sustained reflection and discourse [10]. It is regarded as the core of the CoI, with social and teaching presences required as prerequisites [10, 43].

Garrison et al. [10] proposed four phases—triggering event, exploration, integration, and resolution—that could improve learners' cognitive presence in online courses. During these four phases, learners are allowed to brainstorm, discover, discuss problems, and reflect and integrate ideas during learning activities.

A blended learning course can be challenging to create because it requires instructors, as designers, to carefully and thoughtfully integrate both face-to-face and online content and activities in a way that upholds student interaction, engagement, and meaningful learning. The CoI framework can be applied to effectively structure the development and implementation of a blended learning course wherein collaborative learning is supported. In such a blended learning environment, students can be given opportunities to contribute to collaboration, thereby strengthening social presence. The critical role of teaching presence is emphasized here because collaborative learning and teaching practices are necessary to facilitate meaningful learning.

Because the CoI framework can positively impact students' satisfaction and higher-order learning [49, 50], both the CoI and the collaborative learning approach were considered in designing the BCL course.

## Research questions and definition of terms

Literature describing the teaching materials, teaching methods, forms of interaction, and roles in the class is limited; therefore, creating new and effective models of blended learning that educators can implement or pattern from is difficult [51]. Furthermore, a review of previous research in multicultural classroom settings reveals a lack of studies describing specifically examined constructs related to CoI. More in-depth studies investigating the roles of BCL in establishing a CoI in a multicultural classroom are also required. Whereas the CoI has previously been used to evaluate blended learning modes, learner experiences captured only by class observation and listening to student voices largely represent a new terrain. Through this study, we obtain addition insights into the CoI's applicability in multicultural settings.

The following research questions state the particular aims of this study for examining BCL in multicultural classrooms:

1. How can BCL be integrated in a multicultural classroom?

2. What learnings are derived from the implementation of BCL in a multicultural setting in terms of three interdependent CoI elements—teaching, cognitive, and social presences?

## Methodology

### Study context

Our case study uses a triangulation design in which different but complementary data related to the same topic are obtained by converging quantitative and qualitative methods to best understand the research problem [52]. The study examined a graduate-level blended-format social science course in a southern university in Taiwan between January and May 2020. The course was mainly taught in the English language although the instructor and the Taiwanese students did switch to speaking in Mandarin from time to time. This particular case was selected based on three main criteria: (a) the class is composed of students of different nationalities or cultural backgrounds, (b) the course uses a blended learning approach, and (c) class activities are mainly collaborative in nature.

At the time of the present study, the world was at the brink of a global pandemic. After extending school opening by two weeks following the winter break of school year 2020 as an early response to the pandemic, schools in Taiwan reopened and have continued to remian open even while most of the countries around the world have been implementing school closures [53].

As soon as Taiwan learned about the coronavirus outbreak, the government took action to prevent its widespread transmission and thereby avoided closure of businesses and educational institutions. A two-week extension allowed schools to disinfect educational facilities, distribute medical supplies, and implement new procedures for schools reporting confirmed coronavirus cases [54].

Although in terms of continuity of education the pandemic situation in Taiwan was better than that in other countries, school administrators expressed a desire for university professors to plan ahead in case of a resurgence of new cases. In anticipation of possible future disruption, instructors were asked to adopt, learn, and practice innovative ways of proactively delivering education.

### Participants

Data were collected from the instructor and the ten graduate students—three males and seven females—invited to participate in the study (Table 1). Consent for in-class observation and interviews was given by both the instructor and the students. Because the course was offered under an international graduate program, it used English as its medium of instruction (EMI), although the instructor switched to Mandarin from time to time to explain a particular instruction or concept to the local students. The students enrolled in the class came from Taiwan, Indonesia, and the Philippines, making it a multicultural classroom, and the instructor was a Taiwanese professor.

### Data sources and analysis

**CoI questionnaire.** The CoI questionnaire developed by Arbaugh et al. [55], comprising of 34 items measured on a five-point Likert scale ranging from 1 (*strongly disagree*) to 5 (*strongly agree*), was used as an assessment instrument in the study. The survey instrument included items for evaluating the three CoI elements: teaching presence (13 items), social presence (9 items), and cognitive presence (12 items). The reliability coefficients for each dimension were 0.94, 0.91, and 0.95, respectively.

**Table 1. Profile of the student respondents.**

| Role | Nationality | Gender | Enrollment Status |
|---|---|---|---|
| Ph.D. student | Indonesia | Female | Full-time |
| Ph.D. student | Indonesia | Female | Full-time |
| Ph.D. student | Philippines | Female | Full-time |
| Ph.D. student | Taiwan | Female | Full-time |
| Ph.D. student | Taiwan | Female | Full-time |
| Ph.D. student | Taiwan | Female | Part-time |
| Ph.D. student | Indonesia | Male | Full-time |
| Ph.D. student | Taiwan | Male | Full-time |
| Master's student | Taiwan | Female | Full-time |
| Master's student | Taiwan | Female | Full-time |

**Learning and teaching platforms and sites.** The processes of implementing the graduate course in both face-to-face and online modes were documented, with both physical and online classes observed during the entire 18 weeks of the course. Chat histories in the group chat box of the social media application Line, which was created to maintain communication and to facilitate "housekeeping," along with the content of the collaborative tools used in the class —Google Docs, CyberUniversity, and Google sites—were checked.

**Interviews and reflective essays.** To ascertain their opinions about the course, the instructor and students were asked to participate in semi-structured interviews with open-ended questions mainly based on the CoI framework to allow the researcher to further explore the issues raised by the participants. Each interview lasted for 30–60 minutes; interview data were collected through digital recording and then transcribed into a text document. To ensure that the interviews were accurately transcribed and correctly captured the participants' points of view, interviewees were asked to check the transcripts and approve their statements before the transcripts were analyzed.

The students were also asked to submit reflective essays describing their experience of the blended learning course as well as their insights related to the viability of such a course. Both the interview transcripts and reflective essays were subjected to content analysis, and the recurring words, phrases, and patterns in the transcriptions were categorized into themes and sub-themes.

**Procedures.** Both the instructor and students were informed of the study during the early weeks of the course, and interviews were conducted during the last few weeks of the course by two of the study's researchers. Audio files were transcribed, and together with the course documents and written observations, they were coded by two trained coders.

## Results

### The integration of the blended and collaborative learning in a multicultural classroom

In this section, we address the first research question by describing the structure and details of the BCL course following Neumeier's [9] framework of parameters for the design of blended learning. The framework focuses on specific teaching and learning requirements originally presented under six different parameters; in this study, however, these are combined into four parameters for a more straightforward presentation: (a) *mode and integration*, or the blended learning approach and nature of activities; (b) *teaching method and distribution of learning content*; (c) *involvement of learning subjects*, or the types of interactions taking place; and (d)

*location*, or the physical place where the learning takes place. Together, these parameters were used to describe the BCL design in a multicultural classroom.

**Mode and integration.**   The face-to-face mode, comprising lectures, discussions, and student classroom presentations, was conducted for three hours every week. The online mode, with computer-assisted activities, comprised of meetings via Google Meet, submission of individual assignments via a learning management system called CyberUniversity, and question and answer (Q&A) sessions, collaborative assignments, and peer evaluation of student outputs performed via Google Docs.

The participants conducted 15 out of the 18 meetings of the regular class hours in the face-to-face mode, making it the principal mode in this blended learning course, and the remaining 3 meetings in the online mode (synchronous), not including the time spent on collaborative activities such as Q&A, collaborative assignments, and peer evaluation via Google Docs. Students reported that they spent an average of 1.5 hours per week outside class hours for collaborative activities, resulting in approximately 1,980 minutes spent in the online mode. Time spent on communication and collaboration through the social application Line was also not accounted for in the percentages (Table 2).

The face-to-face and online components of the course were mostly integrated in parallel with one another rather than in sequence. However, overlapping in the implementation of the two modes did occur, especially during certain in-class activities that required small groups of three or four people to work collaboratively on a learning task using Google Docs. The Line group, the CyberUniversity (functioning as a submission bin and repository of student outputs), and the course website based in Google were maintained throughout the semester to keep contact and an active line of communication.

According to the instructor, face-to-face meeting was chosen as the principal mode of the blended learning course to promote more in-person interaction and socialization between local and international students, as management of group dynamics is easier, more flexible, and less structured during in-person sessions. However, the online mode also helped in making the multicultural class function well by presenting opportunities for one-to-one or small-group collaborative learning. According to Gabb [15], managing the group dynamics in a multicultural group of students, centered on measures that determine the psychosocial climate in the classroom, is an important part of a teacher's repertoire. With the integration of the two modes, the instructor of the BCL course was able to implement measures facilitating interpersonal relationships and meaningful learning.

**Teaching method and distribution of learning content.**   A large proportion of the content covered in the course was taken from a seminal work on the diffusion of innovation theory. The content was distributed following the organization of the main textbook, and a complete syllabus containing the study plan for the course was uploaded in the course website on Google Sites.

The course followed a flipped-classroom strategy, wherein the students read the materials and collaborated in online Q&A activities and other collaborative assignments at home and

**Table 2. Blended learning modes.**

|  | Face-to-face mode | Online Mode |
|---|---|---|
| Location | Classroom | Online (videoconferencing platform, online word processors) |
| Time spent for synchronous and asynchronous activities | 180 mins/week | 90–180 mins/session or task |
| Actual contact time (synchronous) | 83% | 17% |
| Estimate of time spent per mode (minutes) | 2,700 mins (58%) | 1,980 mins (42%) |

then explored the concepts under the guidance of the instructor in the physical classroom. During the face-to-face classes, discussions revolved around the questions and answers posted by each student in the Google Docs, with the instructor clarifying misconceptions and elaborating on and summarizing key ideas. Individual and group presentations were also conducted in the face-to-face mode, although not exclusively. Translanguaging, or the use of all of one's linguistic and cognitive resources to teach or learn academic content, was evident during the BCL course. For example, the instructor would use the Mandarin term for the word *innovation* to make it more understandable to local students and then ask the other international students to give examples of an innovation as they understood it in their own context.

Online activities were performed both synchronously and, most of the time, asynchronously. Discussions via the videoconferencing site Google Meet were conducted as a group during regular class hours. Collaborative Q&A activities as well as collaborative assignments such as group memos were conducted asynchronously using the online word processor Google Docs. One student presentation activity was performed online; students pre-recorded their presentations and uploaded them to CyberUniversity, allowing other students to access the presentations and ask questions or give feedback.

When asked about the distribution of learning content and the choice of the blended learning strategy, the instructor explained that she wanted to deviate from traditional lecture methods so that students from different cultural backgrounds could engage in discussions and thereby achieve personal conceptualizations of the course content. She said that using elements of flipped learning, students were able to come to class prepared and ready for teacher and peer feedback on their work.

The teaching methods used in the face-to-face and online modes were different. The face-to-face mode was more flexible in that the instructor could select from a wide variety of and from various combination of teaching methods and approaches, including lectures, inquiry, discussion, group memos, and student presentations. The online mode was limited to the instructor giving feedback on content understanding and some discussion through videoconferencing.

**Involvement of learning subjects.** Individual work, pair work, and group work were forms of communication used in both modes of learning. In creating the small groups or pairs, the instructor tried to group together foreign and local students to ensure a "cultural mix;" however, as the local students outnumbered the foreign students, only a few groups had at least one foreign member. The interactional patterns occurring during the different collaborative class activities usually involved computers—interactions took place either through or at the computer. Table 3 lists the types of interactions and different collaborative activities.

During the course, the instructor mainly acted as a facilitator, a resource, and a collaborator; the students also took on different roles such as recipients, partners, and peer teachers or evaluators. Switches from one role to another depended on the nature of the scheduled activity and were not immediate and dynamic. As the course progressed, the roles became better-defined and familiar among the students. The different types of interaction allowed opportunities for students to interact and work with others using different technological tools.

**Location.** Face-to-face meetings were conducted in a traditional lecture room with synchronous and in-person interaction both between the instructor and the students and among the students. Students brought their personal computers to class, although digital documents and other multimedia materials for class viewing were usually projected for the entire class. During the course, students spent a substantial amount of time on collaborative learning using their computers; during the online mode, this activity could take place in the study room, in the library, at home, or at any location where technology, including internet connectivity, was available.

**Table 3. Some interaction patterns during collaborative activities (adapted from [9, 56]).**

| Type of Interaction | | Collaborative Activities |
|---|---|---|
| Interaction through and with computers/ networks (synchronous or asynchronous) | Instructor to students | Discussion and feedback on Q&A |
| | | Feedback on presentations |
| | | Presentation of supplementary materials |
| | Instructor to a group of students | Feedback on group memo |
| | Student to student | Discussion on Q&A |
| | Group of students to individual student | Peer feedback on pre-recorded presentation |
| | | Peer feedback on final presentation |
| | Student and/or instructor to computer (Google Docs) | Posting two questions in the Q&A |
| | | Modifying and contributing answers to a particular question (asynchronous) |
| | | Peer feedback on a pre-recorded presentation (asynchronous) |
| Interaction at computers/networks | Student and student (or group of students) collaboration at the computer (Google Docs) | Real-time Q&A collaboration (e.g., students A and B discuss the questions posted by student B before student A answers student B's question) |
| | Student and instructor collaboration at the computer (Google Docs and Google Meet) | Posting and responding to comments on the Q&A, or open forum during student presentation in Google Meet |

## Learnings from the BCL experience

Based on the average scores from the CoI survey, the students reported high levels of agreement with items reflecting teaching presence (M = 60.7, SD = 4.19) and cognitive presence (M = 53.6, SD = 4.77). In contrast, the mean score related to social presence (M = 38.4, SD = 4.20) was relatively low compared with those for teaching and cognitive presence.

The findings are presented as four learnings from the BCL experience that emerged from the reflective analysis of the qualitative data generated from class observations and interview transcripts: (1) teaching presence is of central importance, (2) collaborative activities enhance social and cognitive presence, (3) language influences social and cognitive presence, and (4) BCL presents both opportunities and challenges.

**Learning 1: Teaching presence in a BCL environment is of central importance.** The teaching presence in the BCL model was highest among the three presence categories at M = 60.7 or 93%. The interviews suggested that most students recognized the complex and varied roles of the instructor in the BCL model that ranged from communicating the design and organization of the course using available communication channels to facilitating class interactions during face-to-face and online synchronous and asynchronous activities, providing direct instruction by focusing discussion on relevant topics and giving feedback, as well as bridging the language gap in an EMI class with non-native English-speaking students who were not all Mandarin speakers.

All students mentioned that the instructor explained the design and organization of the course at the beginning of the course and prepared a syllabus useful in navigating the BCL course. A student said that the instructor's presence was considered important in holding together the different elements of the "complex course design."

Students said that in terms of facilitation of a multicultural class, the instructor tried to mix everyone as much as possible to enhance social interaction, managed time well during synchronous sessions (whether online or face-to-face) to maximize collaboration and discussion, and used different methods and tools to improve cognitive presence in the class. One student said that the instructor "*gives feedback in a very peaceful manner after first giving encouragement.*"

Regarding providing direct instruction, students mentioned that the instructor gave examples from current events and personal experience, and regularly used the Internet, including video sharing platforms and search engines:

*(S04)*: *"Well, I think the helpful part [is when] the students may put forward different opinions on a certain problem, then the teacher may do a little summary of the concept or generalization after everyone has finished speaking, and then it is just clearer. This is consistent both in online and in the (physical) classroom."*

Outputs related to collaborative activities were useful in increasing teaching presence in the BCL class. For example, the written Q&A in the Google Docs served as a strategic tool for the instructor to provide direct instruction to the students. Because the Q&A was accomplished before a face-to-face class, during the actual class, everybody would look at and discuss it, identifying areas of confusion or incorrect understanding from the way the questions and answers were formulated:

*(S10)*: *"It happens that a reference sometimes provides wrong caption or information and distracts our understanding of the theory, but the instructor clarifies it so it helps us get back on track. Sometimes we have a little bit [of] wrong understanding on the theory, but during the class discussion, it is clarified. Some students misunderstand, and then during the class discussion and explanation of [the] instructor, it becomes clearer."*

The in-class discussion of the posted questions and answers also became a venue for giving and receiving individualized feedback:

*(S03)*: *"Instructor role in this course was not only to address the main concepts of the theory but also to summarize students' answers and give feedback on the student's learning."*

The role of the teaching assistant (TA), who was also a student in the class, was crucial in implementing the BCL approach. The TA provided technical and logistical support to the instructor, freeing the instructor of logistical concerns and allowing her more time to perform other important roles.

**Learning 2: Collaborative and interactive activities enhance social and cognitive presence.** Despite scoring lower than teaching presence, the other two presence categories were rated fairly high by the students, with cognitive presence in class and social presence having mean scores of 53.6 or 89% and 38.4 or 85%, respectively. Our findings showed that BCL activities that encouraged students to work together and interact with one another enhanced social and cognitive presence, as explained below.

The students identified the Q&A activity as the "*most special thing*" about the course. It represented both the collaborative and interactive activity that was recognized by some as a strategy by the instructor to help students learn the topics better; a means to gain different perspectives and feedback and to refine one's own thinking; and a way of discovering the personalities and backgrounds of classmates through the way they constructed and answered the questions. From the instructor's point of view, the Q&A allowed to gauge the students' learning in real time:

*(Instructor)*: *I will be able to identify the concepts that they don't understand or they failed to cover. That would be the challenge that would be faced in other courses as well because of the time constraint. For instance, the class is three hours, but there is a lot to cover. I would be*

*able to identify which concept they grasped by way of how they present themselves, how they answer the questions, or how they failed to bring up concepts that I consider important. I think that would be the best I can do in class. I would be able to assess learning later when I give some assignments.*

Students knew that the questions and answers remained in cloud storage and could be referred to at any future time. Although the written Q&A in Google Docs was learning material, additional learning occured when the questions and answers were discussed during face-to-face sessions.

Discussion of a contentious topic brought to the attention of the class, either by the instructor or by students, is thought of as good training in learning from the perspectives of others and in working hard to check one's own understanding and views. It also challenges students to speak "*rationally*" and "*scientifically*."

*(S07): "I think she encourages us to find resources. To find the support for our agreement or disagreement; if [you] agree on something, what is your support and what is your reference? When we have ideas, we need to cite theory, reference/citation, research findings. . . so that our ideas are strong, you know, not just personal claims. . . and not just our opinion but it should be supported by evidence."*

The Q&A also engendered feelings of excitement and motivation in some students not only because of what they learned from their own Q&A posts but also because of what they gained from looking at others' posts. The Q&A becomes a *stage* where everyone's statements are exposed to all. One student said "*I often go to see other people's writing, which is quite fun.*" Another student said that she was able to form an impression of her classmates because of what they wrote:

*(S09): "For example, (classmate A) will ask about the latest trends, while some will be more serious, asking questions on the education side. You get a clue on what is their background. You know them by the way they ask and answer questions."*

The pre-recorded presentation was another memorable and helpful activity for the students. It was both fun and constructive for the students and was recognized by both the instructor and students as a cognitive experience:

*(Instructor): "When students pre-record their presentations with some time limits, like five minutes, their presentations have better quality than [when they just present] in the physical settings. Because. . . they have the time to rehearse it. And because they are given a time limit of five minutes, they have to pre-plan everything. They can always redo it. It's a kind of "rehearsal, rehearsal, and rehearsal." And during the rehearsal, they are learning much more. It's almost like writing an article in an oral format.*

*And I think it is actually a very rich cognitive learning activity for the students. Sometimes, in the face-to-face setting, [when] I say you only have to do it in five minutes, I feel that students just speculate—"Oh yes, my presentation will end in five minutes." They do not actually do the rehearsal. But when they actually present, it is usually over the five-minutes limit. But if [it is pre-recorded], they know how to organize their presentations. It turns out that they are learning more when they do the presentation online."*

*(S05)*: *"I recorded the presentation many times for the first time. After listening to it, [I thought] how come my English is so bad, and then I record[ed] it again (laughs)."*

**Learning 3: Language influences social and cognitive presence.**   The students attributed the slightly polarized reactions of local and foreign students to the course to language as a multicultural aspect. For one student, this is a more important issue than the blended learning approach to the course:

*(S06)*: *"What I have sorted out is that this may be attributed to language, but the second fact is that they will reflect the difference between face-to-face courses and online courses for them. . . but at the end it is the language problem; that is, Taiwanese students cannot be like foreign students in English. Although they (foreign students) are not native speakers, their English is better than ours, so when we want to express our opinions smoothly, because of language problems, it will be hindered."*

When addressing the entire class, local students used English, whereas some reverted to Mandarin during group work, a recurrence to native language consistent with the findings of other EMI studies [57, 58]. However, during open class discussions, local students tried to speak English as much as possible so that the foreign students could understand them. One student mentioned that the instructor consciously took care of the problem by encouraging local students to speak their own language when they were not able to fully express their opinions using English and then translating what they said for everyone to understand or by suggesting some English vocabulary to capture what they wanted to express. Local students who thought this situation was very troublesome because it "*delays the class*" sometimes chose not to speak and were concerned that their silence might be misconstrued as not "*willing to participate.*"

*(S06)*: *"I believe that most of the Taiwanese students present should also be able to keep up. . . We all know what everyone is talking about, but sometimes we want to interrupt and say something. But I feel that maybe people will wait for me for a long time, and then I feel very embarrassed because my language is not so smooth and fluent; so I am less interested in expressing my own opinions. Otherwise, I actually understand the content of the class and understand everyone. . . there is no problem with the content of the discussion."*

Because the foreign students could better express themselves using the English language than local students and thus had "*richer*" content, the tendency of the class was to wait until a student finished speaking. As a result, only those who could speak faster and more fluently than others dominated the discussion. A student said that local students "*are not unwilling to put forward their own ideas or have no ideas*" but the "*cost of putting forward ideas is too high, and they do not want to delay everyone.*"

Such a situation makes face-to-face discussion even more favorable than online discussion to local students. The use of the videoconferencing platform created pressure to speak faster and not use so much time. Students felt that the time spent in Google Meet was more restricted than the time spent face-to-face and that everybody should make the most of their "airtime," or opportunity to speak. Moreover, the "*embarrassing*" situations were compounded because in Google Meet, where only one person should talk at a time, waiting time becomes even more pronounced.

Whereas feelings of embarrassment and lack of confidence were shared by most local students, one or two of them were outspoken and often contributed during class discussions. In our interviews, we found that these particular students had been more exposed to EMI classes and had study abroad experiences.

**Learning 4: BCL presents both opportunities and challenges.**   Although the online environment afforded some conveniences, such as being more comfortable, "at rest," or less pressured, the constraints of technology use can cause problems in learning and class engagement. Online discussions tend to face more hindrance because during videoconferencing, queuing and regulating who will speak is unavoidable and necessary; students can seldom act spontaneously during online discussions because speaking out of turn or interrupting another speaker can hinder understanding. Therefore, online discussion is deemed by many as not suitable for free and open discussion, especially for interesting topics where many participants want to express their views.

For example, the limited Q&A time can become a hindrance to maximizing the benefits of a particular activity. One student said time constraints resulted in activities often being regulated and controlled, especially when the Q&A discussion was done online, resulting in less discussion and more feedback.

*(S05) This should be more two-way. . . but it becomes a step-by-step process (in the online setting). I am a bit hesitant to interrupt; I actually find it (a Q&A post) very interesting but I don't want to be too rude. In the face-to-face, I can interrupt and I can use a very cute expression to show that I am actually friendly and I am not malicious."*

The facial expressions and body language of students that could help in assuaging fears, promoting friendliness and a sense of community, expressing opinions, or proving a point were also described as "*hidden*" and "*strange*" during online sessions. The unnatural conditions in an online session, such as turned-off cameras, people looking at different things, or people looking uninterested, made the interpretation of messages "*much worse.*" Worries of being misinterpreted in an online setting, such as feeling that others will "*feel challenged by a question,*" were also voiced, and these were cited as a reason for less interaction.

Students mentioned that during online discussions, small group activities were more difficult. They said that they used social media applications such as Line and WhatsApp to collaborate with their group mates, and features such as breakout rooms in videoconferencing platforms were recognized by some students to be potentially useful for small group discussions.

Some students mentioned that not much adaptation was required when switching back and forth from online to face-to-face modes and that regardless of the mode, the students who are not willing to engage in the discussion would tend to find a way to stay out of it by, for example, pretending that they are listening during in-class discussion, and turning off the camera during online discussion.

The BCL approach, with its many components and dimensions, was new to all students, and although there is a general agreement that the approach provided a rich and full experience, students felt that it was "*heavy and loaded,*" "*often extended,*" and time-pressured. The grouping and certain activity instructions, which were intended to increase social interaction, contributed to the difficulty of participating in the course because these were often revised. The evolving nature of the collaborative activities (in terms of mechanics) resulted from the adjustments made by the instructor as she deemed necessary to meet the changing needs of the class.

Overall, most students said that although they faced some issues during the BCL course, they found it acceptable and tolerable as first-time students of such a multicultural blended learning class. They believed that online class is a "*very good thing*" because "*this thing needs to be tried*" with or without a pandemic. Although using the BCL approach necessitated both the instructor and the students to "*work very hard*," the two learning and teaching modes, when used properly and according to the planned purpose, could result in a rich and full experience.

## Discussion

The results of this study showed that integrating blended and collaborative learning in a multicultural classroom is not a simple process; it requires the instructor to redesign the course in the form of a flipped classroom, and certain challenges brought about by student-centered learning and multiculturalism need to be overcome.

Because redesigning the course using the flipped classroom model requires instructors to invest time and effort [59, 60] and to exercise good pedagogical judgement, this may be more challenging for less-experienced or novice teachers [60]. Implementing a flipped classroom model also calls for a paradigm shift among teachers from the assumption that students are empty vessels passively absorbing information from the teacher [61] toward a classroom centered around students who have a major responsibility for their own learning. The instructor role is changed from "sage of the stage" or "presenter of content" to "guide on the side" or "coach" [60].

Note that students scored teaching presence in the BCL class as relatively high, reflecting strong facilitation and direction by the instructor in both the online and face-to-face modes of the class. High teaching presence is a positive aspect in BCL in a multicultural classroom, consistent with the findings that effective instructor techniques in multicultural classrooms include setting up the class, interacting with students, providing feedback, and recognizing and valuing cultural differences [62].

Integrating the elements of flipped learning into blended course design appeared to be effective in allowing a broader coverage of the topics in the course. For example, by designing course content such that it can be processed outside of class time, other activities can be accommodated during face-to-face or synchronous online sessions, although some students pointed out that flipped learning reduced some of their personal time and could sometimes be burdensome. Therefore, whereas flipped learning is conducive for student-centered learning theories such as collaborative learning [63], low levels of self-regulated behaviors by some students [64] and poor time management in seeking to comprehend learning content outside class hours [65] may be stumbling blocks for its potential as a viable method for multicultural classroom teaching.

Social presence survey items were ranked lowest among the three presence categories, indicating that not all students were comfortable participating in discussions and in interacting with other course participants. Because all students freely contributed to the Q&A activities using the interactive Google docs, the lower scores of the social presence survey items could be interpreted as representing the participants' higher discomfort with social interaction than with cognitive interaction. However, the literature indicates that social presence should be stronger in a BCL course because it offers a higher levels of interaction than commonly experienced in a face-to-face class owing to the availability of technological tools and platforms that create a communication environment conducive for enhanced learning [66, 67].

A language barrier between local students and foreign students—more fluent speakers of English in class—could be considered a factor contributing to the relatively low social presence in the course [11]. Moreover, collaborative and interactive activities may not be beneficial to

or may create undesirable experiences for those belonging to a particular culture not only because of the language gap but also because of their cultural dispositions (e.g., politeness and preference to not inconvenience others). Because constraints such as time and technology limitations may aggravate the language problem of most local students, addressing these issues is important. This study's findings, therefore, show that cultural aspects may influence the desirability of BCL.

Finally, note that an important finding of this study is related to the effectiveness of a metacognitive approach to the blended learning design. Metacognition includes a critical awareness of one's thinking and learning and of oneself both as a thinker and a learner who plans, monitors, and assesses one's understanding and performance [68]. The pre-recording of presentations was an activity that helped improve metacognition. By giving explicit instructions for the preparation of presentations, the instructor trained the students to make judgements regarding which key messages they would like to highlight in their presentation, to rehearse their report, to remove unnecessary ideas, and to optimally use the limited time provided. In short, by having opportunities to reflect on their class outputs, the students learned how to recognize their own cognitive improvement.

## Implications and conclusion

The findings of this study provide implications for both school administrators and instructors of educational institutions aiming for internationalization.

For administrators, the findings reiterate the importance of supporting and enabling the planning and preparation of BCL classrooms for culturally diverse graduate students. With the widespread use and unavoidable needs for online learning, classes must be equipped with appropriate technological support, including technological infrastructure and accessibility to online platforms. Encouraging instructors and providing them with professional development activities to aid them in facilitating a BCL approach in a student-diverse setting is necessary.

The results of this study can likewise inform instructors aiming to transform their curriculum to create a CoI within their multicultural classrooms. The BCL approach gives instructors an opportunity to experience a variety of instructional strategies and may extend their instructional design capabilities. As social presence continues to be a challenge in blended learning and multicultural environments, relationship maintenance should be emphasized when adopting an online collaborative approach.

Because learning situations differ, and there is no single formula for creating an ideal BCL course, planning for a good BCL approach wherein the online and face-to-face components do not overpower one another could increase the acceptability of the entire BCL approach not just that of one of its components. Future research could also examine how to provide support for enhancing culturally diverse students' social presence using the BCL approach.

This study is subject to limitations. First, although the aim of this study was to demonstrate how a BCL approach is integrated in a multicultural classroom and identify the lessons that can be learned from the implementation of BCL, as a single case study, the results may have limited explanatory range and generalizability—a limitation that is usually tied to a broader critique on qualitative methods as a whole. Second, the participants of the course were mostly Asian students; a case of a BCL class with more culturally diverse students—particularly one that includes Western students—can demonstrated stronger representativeness and provide a broader spectrum of cultural differences. Finally, the study has limited quantitative findings to support the claim that BCL is a viable approach for multicultural classrooms; hence, further research is recommended.

## Supporting information

**S1 Data. Survey data_CoI questionnaire.**
(XLSM)

**S1 File. CoI questionnaire.**
(DOC)

**S2 File. Interview data.**
(DOCX)

**S3 File. Interview details and questionnaire.**
(DOCX)

## Author Contributions

**Conceptualization:** Hsueh-Hua Chuang.

**Data curation:** Aurora V. Lacaste, Ming-Min Cheng, Hsueh-Hua Chuang.

**Formal analysis:** Aurora V. Lacaste, Ming-Min Cheng.

**Funding acquisition:** Hsueh-Hua Chuang.

**Methodology:** Aurora V. Lacaste, Ming-Min Cheng, Hsueh-Hua Chuang.

**Project administration:** Hsueh-Hua Chuang.

**Supervision:** Hsueh-Hua Chuang.

**Writing – original draft:** Aurora V. Lacaste, Ming-Min Cheng.

**Writing – review & editing:** Hsueh-Hua Chuang.

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
