## [Decision Letter · Decision Letter 0]

16 Dec 2021

PONE-D-21-31362Blended and collaborative learning: Case of a multicultural graduate classroom in TaiwanPLOS ONE

Dear Dr. Chuang,

Thank you for submitting your manuscript to PLOS ONE. After careful consideration, we feel that it has merit but does not fully meet PLOS ONE’s publication criteria as it currently stands. Therefore, we invite you to submit a revised version of the manuscript that addresses the points raised during the review process.

We look forward to receiving your revised manuscript.

Kind regards,

Fu Lee Wang

Academic Editor

PLOS ONE

Journal Requirements:

Reviewers' comments:

Reviewer's Responses to Questions

**Comments to the Author**

1. Is the manuscript technically sound, and do the data support the conclusions?

Reviewer #1: Yes

Reviewer #2: Yes

2. Has the statistical analysis been performed appropriately and rigorously? 

Reviewer #1: N/A

Reviewer #2: N/A

3. Have the authors made all data underlying the findings in their manuscript fully available?

Reviewer #1: Yes

Reviewer #2: Yes

4. Is the manuscript presented in an intelligible fashion and written in standard English?

Reviewer #1: Yes

Reviewer #2: Yes

5. Review Comments to the Author

Reviewer #1: This study presents a blended and collaborative learning method and discuss its potential as a viable learning approach for graduate classrooms with culturally-diverse students. The topic is interesting. In my view, the findings of this study can provide important insights into instructional practice on blended and collaborative learning. I only have several minor concerns to be addressed.

First, I suggest the authors to highlight more about the importance and significance of blended learning and collaborative learning in the field of educational technology, as evidenced in reviews on educational technology, see the followings:

Detecting latent topics and trends in educational technologies over four decades using structural topic modeling: A retrospective of all volumes of Computers & Education

Fifty years of British Journal of Educational Technology: A topic modeling based bibliometric perspective

It is suggested to include more empirical studies focusing on blended and collaborative learning to highlight the significance of your research. For example,

Kong, S. C., Cheung, W. M. Y., & Zhang, G. (2021). Evaluation of an artificial intelligence literacy course for university students with diverse study backgrounds. Computers and Education: Artificial Intelligence, 100026.

Research questions can be more specific, for example, RQ2, what does “knowledge” means?

What are the limitations of this study?

Finally, double-check both definition and usage of acronyms: every acronym should be defined only once (at the first occurrence) and always used afterwards (except for the abstract). There are some mistakes in this aspect. For example, the use of CoI and community of inquiry.

Reviewer #2: This paper describes an implementation of a blended and collaborative learning approach in a multicultural graduate course, intending to demonstrate such an implementation can encourage cooperative learning among graduate students of different cultural backgrounds. The paper evaluates the learning outcomes using the Community of Inquiry framework. In general, the paper delivers a teaching and learning event with a theoretical foundation and a well-constructed learning process. Thus, the paper is of interest to the journal and researchers in the relevant field.

The authors can consider the following suggestions:

1. Organization:

1) It will be better if the authors number the sections and subsections. The current format without numbering may make the audience get lost.

2) Line 288. It is confusing to entitle this section as "Results," as it is more about the teaching and learning settings and some basic statistics. The authors can consider sperate this section accordingly to make the manuscript logically intact.

3) Another limitation of the manuscript is that the description of blended and collaborative learning in the current "Results" section is not well-written: the content is too scattered without a good connection, and the writing is not good enough to present a holistic image. The authors are highly suggested modifying this section to make everything straightforward. For example, authors can consider writing a general description to start this section, highlighting the logic and how different subsections are connected.

4) The "Results" part does not discuss how this work addresses from a multicultural perspective and impacts the learning of students from minor ethical groups, which is emphasized a lot in the abstract and introduction. Please consider adding discussions regarding the multicultural setting into the manuscript.

As to the claim about multicultural setting, more literature can be added to support the hypothesis. And is there any quantitative results to reveal the effect of the proposed BCL setting on both local students and foreign students? I think it would be interesting if these aspects are discussed.

2. Writing:

The writing quality of this paper can be improved. The authors are suggested to make thorough proofreading.

Typos:

Line 100: "equity" to "equality"

Line 133: "others" to "others' "

Line 140: "The" to "the"

Line 298: "exercisew" to "exercises"

and other places.

Especially, the paper does not well follow the citation format. For example, multiple in-line citations miss the period sign "." in "et al."

As to the use of acronyms, once a short form is defined, please use the acronym throughout the manuscript, like "BCL". The term "Community of Inquiry" is defined as "CoI", but another form "COI" is used multiple times. Please stick with one form and revise the other one. And "Q and A" may not be a formal usage. Please consider using "Q&A" instead.

6. PLOS authors have the option to publish the peer review history of their article (what does this mean?). If published, this will include your full peer review and any attached files.

Reviewer #1: No

Reviewer #2: No

---

## [Author Response · Author response to Decision Letter 0]

2 Mar 2022

Dear Dr. Fu-Lee Wang,

We would like to submit the revised manuscript, “Blended and collaborative learning: Case of a multicultural graduate classroom in Taiwan” for possible publication in PLOS ONE. We appreciate the insightful comments from you and reviewers and we have used them to revise the article into what we believe will contribute to the profession.

We have also provided a detailed response letter for each of the reviewers, according to reviewer’s comments, to highlight the significance of the study and strengthen the research rationale by adding more empirical studies focusing on blended learning and collaborative learning, providing discussion regarding the multicultural setting. Additionally, we have added limitations in the implications and conclusion section to provide potential factors that might influence the study outcomes. In addition, we have numbered the headings in compliance with the suggestion of one the reviewers. We have listed the reviewer’s comments and then carefully addressed each of them in the responses. All revisions in the manuscript are marked up using the “Track Changes” function.

Funding: This work was supported by Office of Higher Education Sprout Project 2020 from Ministry of Education in Taiwan. (no grant number available). The funder had no role in the study design, data collection and analysis, decision to publish, or preparation of the manuscript.

Thank you for your consideration for possible publication opportunities. Please let us know if you need additional information.

---

## [Decision Letter · Decision Letter 1]

14 Apr 2022

Blended and collaborative learning: Case of a multicultural graduate classroom in Taiwan

PONE-D-21-31362R1

Dear Dr. Chuang,

We’re pleased to inform you that your manuscript has been judged scientifically suitable for publication and will be formally accepted for publication once it meets all outstanding technical requirements.

Kind regards,

Fu Lee Wang

Academic Editor

PLOS ONE

Additional Editor Comments (optional):

Reviewers' comments:

Reviewer's Responses to Questions

**Comments to the Author**

1. If the authors have adequately addressed your comments raised in a previous round of review and you feel that this manuscript is now acceptable for publication, you may indicate that here to bypass the “Comments to the Author” section, enter your conflict of interest statement in the “Confidential to Editor” section, and submit your "Accept" recommendation.

Reviewer #1: All comments have been addressed

Reviewer #2: All comments have been addressed

2. Is the manuscript technically sound, and do the data support the conclusions?

Reviewer #1: Yes

Reviewer #2: Yes

3. Has the statistical analysis been performed appropriately and rigorously? 

Reviewer #1: Yes

Reviewer #2: Yes

4. Have the authors made all data underlying the findings in their manuscript fully available?

Reviewer #1: No

Reviewer #2: Yes

5. Is the manuscript presented in an intelligible fashion and written in standard English?

Reviewer #1: Yes

Reviewer #2: Yes

6. Review Comments to the Author

Reviewer #1: The authors have addressed all my previous concerns. I feel the current version can be accepted for publication.

Reviewer #2: I would like to thank the authors for their effort on revising the manuscript and addressing my concerns. The revised version shows significant improvement compared with the previous version. The current version can be considered for publication.

7. PLOS authors have the option to publish the peer review history of their article (what does this mean?). If published, this will include your full peer review and any attached files.

Reviewer #1: No

Reviewer #2: No

---

## [Editor Report · Acceptance letter]

19 Apr 2022

PONE-D-21-31362R1 

Blended and collaborative learning: Case of a multicultural graduate classroom in Taiwan 

Dear Dr. Chuang:

I'm pleased to inform you that your manuscript has been deemed suitable for publication in PLOS ONE. Congratulations! Your manuscript is now with our production department. 

Kind regards, 

on behalf of

Professor Fu Lee Wang 

Academic Editor

PLOS ONE